# Muslim YouTubers in Turkey and the Authoritarian Male Gaze on YouTube

**Esma Çelebioğlu**

Department of Cultural Studies, George Mason University, Fairfax, VA 22030, USA; hcelebio@gmu.edu

**Abstract:** The increased digitalization in today's world, including social interactions online, as well as digital practices and performances, has a significant impact on the identity formation of youth and reflects their self-representation in society and the global world. This article examines how gender identities shape online representations of religious youth in Turkey. To this end, this study particularly focuses on young Muslim YouTubers whose religious identity appears either as a part of their images (i.e., veiled women/hijabi YouTubers) or through the contents they create (Muslim male YouTubers). Presenting similarities and divergences between online representations of Muslim YouTubers, this study sheds light on how Muslim youth express religiosity as a part of their online identities through the digital content they create. Furthermore, this analysis explores different modes of utilization of YouTube by young female and male Muslims as manifested through their videos. Following Schiffer's categorization of the functionality of objects, I will argue that socialization and individuality are highly prioritized in the contents created by hijabi YouTubers. At the same time, an ideological and authoritarian perspective becomes prominent among the YouTube videos created by Muslim male YouTubers in Turkey.

**Keywords:** digital media platforms; techno-religion; digital religiosity; muslim youtubers; gender identity; religious authorities

## 1. Introduction

Digital media and online platforms are constantly reconfigured and transformed through the online performances and digital practices of net users. The interactions of individuals with digital media also play an influential role in their self-identification process, simultaneously redesigning their social interactions and relations with the offline world. New generations, being born into this digitalized world, are increasingly engaged in digital platforms and utilize digital media technologies much more actively than past generations (Ito 2009). In this respect, both youth and younger generations (kids) participate in the digital world not only as users but also as makers by creating "their own meanings and media products" (Ito 2009, p. 13). The increasing utilization of digital media has a crucial impact on the identity formation of this new generation—*netizens*—and reshapes their self-perception and self-positioning in society and the global world (Boyd 2008; Liu 2011).

The rising popularity and utilization of digital media platforms are also significant in shaping the lifestyles of Muslims, particularly playing an influential role in the identity construction and socialization of Muslim youth (Kavakci and Kraeplin 2017). Lately, we have witnessed that young Muslim women and men worldwide have become much more visible on digital platforms by producing numerous content and displaying different types of online identities. Online visibility of Muslim identities and their digital performances plays an increasingly significant role in shaping the religious identities of Muslim youth through their engagement in diverse styles and trends in the global world (Kavakci and Kraeplin 2017; Baulch and Pramiyanti 2019).

This article examines the interplay between digitalization and the identity formation of Muslim youth in Turkey based on a small-scale analysis of the contents created by

Muslim YouTubers and their digital performances. This study does not aim to offer a systematic overall analysis, nor does it claim to be representative of a defined community. Instead, the primary goal of this study is to present a snapshot of the online identities of Muslim YouTubers in Turkey through case study analysis. Drawing its findings from a virtual ethnography based on discourse analysis, this study mainly aims to answer these research questions:

- How do Muslim youth in Turkey display their religious orientation in their online performances, and how does religiosity appear in the online identities of Muslim YouTubers in Turkey?
- In what ways do gender roles and identities shape online self-representations of Muslim YouTubers in Turkey?

Accordingly, following Schiffer's categorization of the functionality of objects, different utilizations of YouTube by Muslim YouTubers in Turkey are discussed. The study argues that socialization and individuality are much more prioritized in the contents created by hijabi YouTubers, while an ideological and authoritarian perspective becomes prominent among YouTube channels and videos created by young Muslim male YouTubers in Turkey.

## 2. Digitalization of Religion and Islam in Digital Media

Even though religions and belief systems are usually considered based on their metaphysical aspect, religious systems are closely connected to the offline world since they are conceived, experienced, and performed by individuals within this material world. As a system of belief, religion regulates and organizes the practices of individuals, their lifestyles, and their social relations. In this regard, it comprises different forms of ideas, sets of belief, ritual practices, and performances in its dynamic and heterogeneous structure. Belief systems, the conceptualization of religions, and religious practices have also coevolved with other social structures (Taylor 2007). In this framework, religious understanding, rituals, and the manifestation of religiosity in today's world have also been altered through its interaction with technology. The integration of technology into religion inevitably transforms practicing religion and creates new versions of religious performances (Campbell 2004; Possamai and Turner 2012; Wagner 2013; Campbell et al. 2014; Han 2016; Bunt 2018).

Today, we can talk about the digitalization of religion in many ways. There are multiple forms of techno-religious devices, various apps for smartphones, and many other digital objects designed and produced for religious purposes (Gorman 2009; Wagner 2013; Campbell et al. 2014; Bağlı 2015). There are also numerous religion-based websites and religion-based content on digital platforms, either providing information or enabling interactive communication and social interaction (Helland 2000). In addition to different techno-religious objects that mediate religious practices and performances, people can share and exchange their ideas on digital channels and even participate in online forms of religious performances (Possamai and Turner 2012).

The implementation of digitalization in religion encourages and promotes the visibility of religious actors, content, and religious performances on digital platforms. Most religious organizations and famous religious figures have their websites, official and unofficial Facebook pages, Instagram accounts, YouTube channels, and blogs. At the same time, by opening up more space for the debate and discussion of religious topics in which alternative voices, different perspectives, and interpretations can be easily and freely shared, digital media prompts the emergence of new religious figures, bringing on "a multiplication of competing authorities" (Possamai and Turner 2012, p. 200). By enabling mass participatory communication, digital media platforms allow individuals an interactive and dynamic arena where people can discuss, debate, question, and be informed about religious topics.

The integration of digital culture and religious practices shape and facilitate the role of religiosity in everyday life. The articulation of religiosity into modern lifestyles promoted by the digitalization of religion transforms religious performances and religious perspectives. More crucially, they take place in the identity formation of religious people, particularly youth, through an interactive and dynamic relationship with other social structures.

Like other religious systems, Islam also interacts with the developments in the world, such as digitalization, globalization, and consumerism. The digitalization of religion has influenced and transformed Islam/Islamic discourse in numerous ways through the engagement of the Muslim population in the digital world. Today, multiple techno-religious objects have been introduced into the Islamic religious market. Additionally, different religious practices have emerged on digital platforms such as online jamaats, virtual dhikrs, and even online participation in Islamic tariqas (Piraino 2016). In *Hashtag Islam: How Cyber-Islamic Environments are Transforming Religious Authority*, Bunt (2018) provides a meticulous analysis of different utilizations of digital media and technology by Muslim users. While indicating the practical utilization of technology, Bunt (2018) underlines ethical, social, and political outcomes of the interaction of Islamic knowledge, Muslim societies, and technology. As he demonstrates, the close link between the market economy and technology has also reshaped the transformation of Islamic culture. Technology alters Muslim lifestyles with its integration into Islamic markets through a dynamic and interactive relation between economy and technology: "The result is an array of apps that can affect everyday religiosity, observation of precepts, and expression of Islamic identity—from relationships to eating habits" (Bunt 2018, p. 33).

The emergence of new religious figures with the Internet utilization by Muslims also contributes to the remediation and challenge of "hegemonic interpretations of established religious authorities" (Echchaibi 2013, p. 191). Increasing engagement of Muslim individuals in technology and digital media makes them "actors" in terms of shaping civil society, as "new media technologies create a new Muslim public sphere" (Echchaibi 2013, p. 191).

Consequently, online and offline religious identities and religious communities are being challenged, altered, and reshaped within the close and dynamic relationship between globalization, technology, and religion. Creative and innovative as it can be, this relatively new coupling of Islam and technology is challenging by stimulating critical and ethical discussions about the compatibility of these digitalized performances with taken-for-granted Islamic norms.

### 3. Methodology

The methodological approach followed in this study is netnography which is "the ethnographic study of online cultures and communities" (Kozinets 2011). Building primary data upon online sources, this research employs a virtual ethnography based on a discourse analysis of the YouTube videos posted by young Muslim YouTubers in Turkey. By conducting online research in this study, I will examine different YouTube content created by young Muslim YouTubers. It is possible to identify these YouTubers as religious Muslims since their religious identity either appears as a part of their images (i.e., veiled women/hijabi YouTubers) or their YouTube contents reflect their religious personae as Muslim and religiously oriented individuals (Muslim male/religious YouTubers).

To this end, ten different Muslim female YouTubers' channels were reviewed (Table 1). In this process, it was observed that most of these YouTubers share similar themes and content in their videos and follow a similar approach with other hijabi influencers in other parts of the world. Therefore, after this review, three of them were identified and selected for discourse analysis, respectively, Rimel Askina, Aslı Afşaroğlu, and Sule Celik. Three main criteria were considered for selection: age, popularity, and activity on YouTube (number of followers and videos) and the variety of topics covered in YouTube videos. Although there are some other Muslim female YouTubers prominently featuring religious content, they were excluded from this research since they are much older than the religious youth examined in this research.

Secondly, seven YouTube channels designed and organized by young Muslim men or male groups were reviewed (Table 2). These channels are either introduced as Islamic channels or feature religious content. Two Islamic YouTube channels, having more followers with a vast number of videos, namely Sözler Köşkü (a channel organized by a group of Muslim males) and Hayalhanem (Mehmet Aydın), have been selected for further discourse

analysis. The videos and content were carefully reviewed and selected to cover different topics and themes as much as possible. These videos were viewed and examined in detail between September and December 2021.

**Table 1.** Hijabi YouTubers in Turkey.

| Channel | Subscribers | Videos | Joined on |
|---|---|---|---|
| Rimel Askina | 451K | 128 | 3 July 2014 |
| Aslı Afşaroğlu | 349K | 322 | 8 September 2015 |
| Şule Çelik | 78K | 34 | 21 December 2017 |
| Sips Ecmel | 148K | 27 | 22 June 2018 |
| Hatice Ebrar Sağlam | 91.5K | 566 | 3 March 2020 |
| Ask-ı Muhabbet | 76.9K | 121 | 29 July 2020 |
| Mekselina Islam | 74K | 88 | 10 January 2021 |
| Rabia Başoğlu | 34.6K | 36 | 28 August 2015 |
| Sena Pektemek | 12.2K | 14 | 24 September 2021 |
| Derya'nın Kanalı | 2.86K | 25 | 16 January 2013 |

**Table 2.** Islamic YouTube channels and Muslim male YouTubers in Turkey.

| Channel | Subscribers | Videos | Joined on |
|---|---|---|---|
| Sözler Köşkü | 2.68M | 469 | 6 March 2011 |
| Hayalhanem Mersin | 2.48M | 760 | 28 May 2013 |
| Çınaraltı | 624K | 685 | 20 April 2012 |
| Mehmet Yıldız | 412K | 364 | 20 February 2015 |
| Mesken | 332K | 332 | 23 September 2018 |
| Son Liman | 326K | 269 | 2 March 2019 |
| Fatih Yağcı | 197K | 192 | 3 May 2014 |

## 4. Limitations

The findings of this study have to be seen in light of some limitations. The primary limitation to the generalization of these results is the sample size. Many hijabi women in Turkey have recently become popular on different digital platforms such as Instagram, Twitter, Facebook, and YouTube. These Muslim women post digital content on various topics, from fashion to feminist critiques. While the online presence of Muslim youth in Turkey is currently increasing on different platforms, the purpose of this study is to analyze the online presence of Muslim influencers on YouTube, which limits the size of this sample. After examining different performers' activities and videos on YouTube, the hijabi YouTubers analyzed in this study were selected based on their popularity, activities, age range, and YouTube content.

Another limitation occurred in the identification process of Muslim YouTubers. Since the hijab is acknowledged as a religious code for Muslim women, identifying Muslim female YouTubers in Turkey is relatively more straightforward than recognizing Muslim male YouTubers. Since the expression of religiosity in online performances is one of the main objectives of this study, Muslim male YouTubers who specifically produce religious content were included in this study and examined. Many Muslim male YouTubers in Turkey may create a variety of content; however, since religiosity is not manifested in their performances or content, they were not included in this research.

Further extensive analysis with a larger sample size provides a better understanding of the relationship between digitalization and the identity formation of Muslim youth in Turkey. However, it remains beyond the scope of this study and might better be examined in an extended study period.

## 5. Results and Discussion

*5.1. Hijabi YouTubers in Turkey*

The present analysis mainly focuses on three female YouTubers, namely Rimel Askina, Aslı Afşaroğlu, and Sule Celik. Even though many other hijabi YouTubers have recently become popular in Turkey, this study specifically focuses on these three YouTubers to show different utilizations of YouTube by young Muslim females in Turkey. These YouTubers were mainly selected based on their age and popularity. They are all between 20–30 years old and popular on YouTube based on their number of subscribers. As detailed below, Rimel Askina and Aslı Afşaroğlu can be considered professional YouTubers as they mostly build their careers upon their digital performances. Despite being less popular than Rimel Askina and Aslı, Sule Celik was included in this study since her YouTube content is partly differentiated from other examples reviewed in this research. For the content analysis, 15 videos from each channel were picked according to their titles and examined in detail.

### 5.1.1. Rimel Askina

Rimel Askina was one of the first and is one of the most popular hijabi YouTubers in Turkey. Hilal (the owner of the YouTube channel Rimel Askina) is a young woman in her mid-twenties who has been active on YouTube since 2014, with more than 450,000 followers. By January 2022, there were 123 videos posted on her channel that had been viewed more than 50 million times. The first videos of Hilal (Rimel Askina) were mainly about fashion trends, hijab styles, and makeup techniques and tips. In her videos, Rimel Askina mostly talks about fashion styles and trends or introduces makeup brands and beauty products she uses. She also posts videos on different content such as episodes of her fashion shoots, shopping experiences, house design, room decoration, as well as her daily life practices and moments from her social life.

Sometimes, in her Q&A videos, Hilal also mentions her religious experiences and gives suggestions to young Muslim women who want to start wearing headscarves. She even shares her personal religious experiences, such as how she and her close friends consult the Qur'an by randomly picking up a verse or page from the holy book or how she had decided to cover her hair (Rimel Askina 2018). From these videos and comments posted on this channel, we understand that many young women have been inspired by her videos and started to wear headscarves.

She also received many criticisms for using the hijab as a fashionable item. In this sense, she was mostly criticized for her previous hijab style, which did not fully cover her hair. Later, in another video, she shared an email from one of her followers and explained how she decided to cover her hair fully (Rimel Askina 2019). These criticisms and comments are significant to show the impact of visuality in the digital performances of online identities and how it is understood and reflected in different ways by the viewers. Nevertheless, receiving both criticism and praise, she is still one of the most popular and influential hijabi YouTubers in Turkey who encourages and promotes the visibility of Muslim women on digital platforms as content makers.

### 5.1.2. Aslı Afşaroğlu

Aslı Afşaroğlu is another young popular hijabi YouTuber in Turkey. She started her career as a YouTuber in 2015 by posting short sewing tutorial videos. By January 2022, her Turkish YouTube channel, which she refers to as her digital diary, had 347,000 subscribers. There are 328 videos posted on her channel, covering various topics; collectively, they have been viewed more than 36 million times. She also opened an English YouTube channel in 2017; however, there were only three videos in this channel during the writing period of this article. Although Aslı still provides sewing tips, her latest videos are more personalized. As a vlogger, she now shares videos about her travels, favorite makeup brands, clothing choices, books she reads, and details about her shopping practices, decoration, and study practices. After becoming more popular on YouTube, she also created her clothing brand and started to sell her designs online. Although Aslı talks about similar topics with Hilal

(Rimel Askina), such as fashion or shopping, her videos focus more on her daily routines and provide more details about her life.

### 5.1.3. Sule Celik

The last Muslim female YouTuber I would like to introduce here is Şule, who joined YouTube in 2017 with her channel named tıbbiyeliblog, then changed to Sule Celik. This channel differs from the previous ones, both in style and approach. Şule is a young, married woman in her mid-twenties. Both she and her husband are medical students at a university. Şule is much less popular than Rimel Askina or Aslı, with 33 videos on her channel and 77,900 followers.

While Şule's channel is also designed more likely as a vlog or digital diary, the contents she shares in her videos are more focused on their experiences as medical students. Along with these videos about medical education, Şule also posts videos about their studying practices, their daily schedules on Ramadan, and different hijab styles. She also shares a few advertising videos in which she introduces products she uses and suggests to her followers. Recently she also started to add videos about her pregnancy period, giving details and advice to young women about a healthy pregnancy.

### 5.1.4. Online Performances of Hijabi YouTubers

Although the videos posted by these YouTubers show differences in style and the content they share, the most prominent commonality in these videos is the active presence of hijabi Muslim women in the social and public arena and the digital world. The integration of their religious identities into modern lifestyles is also a remarkable point manifested through their videos and channels. In these videos, we mostly witness the integration of the Islamic way of life into a modern lifestyle. These young women are very open about their personal life, and they often share details about their different experiences with their followers. However, none of these YouTubers overtly put forward their religious identity in their content, and none of these videos directly remarks on religious topics. Instead, being a part of their identity, they either share their religious experiences in everyday life or talk about their religious perspective in a manner of dialogue or conversation.

Online self-representations of these hijabi YouTubers and their digital performances are first and foremost significant in the sense of drawing the image of liberated Muslim women. The appearance of hijabi YouTubers on digital platforms with their alternative fashion styles also responds to the negative stereotypical understanding of veiled Muslim women as "the traditional, the backward, the obsolete" (Çinar 2005, p. 62). The fashionable images of hijabi influencers challenge the traditional, patriarchal interpretation of women within the Islamic discourse advocated mainly by male religious authorities. Through developing creative and alternative fashion styles, hijabi YouTubers draw a modern portrayal of Muslim women and aim to create "a progressive, modernist, visibly Muslim aesthetic, with its own visual vocabulary and material forms" (Tarlo 2010, p. 8).

Visuality is a crucial element of digital media that has recently become much more prominent with the proliferation of digital platforms in which visual images and written texts can be posted together. In this context, image-sharing, combined with styles and fashion trends, has significantly reframed the online self-representation practices and digital performances of influencers. Personal images, styles, and performances created and shared via digital platforms define a new lifestyle and a professional career by which people can make money through their online identities and digital performances. Performances of online identities, coalesced around visual images and written texts, play an influential role in the identity formation process of youth through their active engagement in digital media technologies both as creators and followers (audience).

However, digital sites such as YouTube and Instagram play a double role as a mirror by prioritizing visual performances. Digital performances of women on these platforms demonstrate that they can design and control their self-representation. However, they are simultaneously being positioned and viewed as objects of the spectators' gaze.

> While it reproduces the mirror's panoptic logic and the related duty that weighs on them to work on their appearance in order not to be denied their female identity, it is also a possible space of articulation of a female voice on appearance, by and for women, a space for the expression of other images of the fashionable. (Rocamora 2011, p. 422)

Online self-representation of hijabi women, while expressing their religious identity, concomitantly defines them as consumer subjects. As modern individuals adapted to the global world, "hijabers adopt the tactics of microcelebrity culture to posit their consuming bodies as those capable of authoritatively addressing Muslimah in the interpretation of scripture" (Baulch and Pramiyanti 2019, p. 267).

5.1.5. Consumer Culture and Digitalization in Turkey

Consumerism is a salient point of critique through which Islamic authorities address both hijabi YouTubers and the marketization of religion. However, despite criticisms against the threats of consumer culture and modernization, Muslim societies are not immune to the changes in the global world, and consumer culture spreads in these geographies as well.

The visibility of hijabi online identities reveals the economic mobilization of Muslim women as well as their economic and social liberation. However, their online identities also become objects of consumption through their digital performances. Online performances and the self-representation of the online identities of Muslim youth crystallize the impact of consumer culture, neoliberal logic, and globalization on the identity formation process of the youth and the manner in which it is fostered through digital culture.

In most of the videos posted by hijabi YouTubers in Turkey, the effect of consumer culture is remarkable as a part of self-formation, being also visible in the content of many other YouTubers worldwide. As Navaro-Yashin mentions, this is closely related to identity politics expressed through consumption. In other words, consumption forms turn out to be the expression of identities: "A politics of identity had become a politics over symbols in the context of consumerism" (Navaro-Yashin 2002, p. 110). Consumption, in this respect, becomes a leitmotif that defines our subjectivity and identity in today's world (Bauman 2007). Hence, identities become commodities in today's societies with the expanding notion of consumption that exceeds every aspect of life (Goodman and Cohen 2004, p. 37). Trends in the global world, consumer societies, and increasing engagement in digital culture have altogether shaped and altered the identity formation of the youth.

The different performances and styles of hijabi YouTubers epitomize the increasing individualization and fragmentation of religious experiences in contemporary societies. Today's religious identities are not "performed and mediated" as in the past, which had limited sources and was reshaped around localized religious norms (Lövheim 2013). Digital media promotes the emergence of more authentic and alternative religious identities. These identities are interactively informed by local, global, and digital norms and a practice such as "the Internet enhances the possibility of individually practiced religion, but digital media also make visible and provide a new form of social infrastructure for the individual's religion: a network of local communities" (Lövheim 2013). Digital media allow individuals to represent and share their affinities, interests, and even social and political orientations with others. It enables self-representations through digital performances and fosters social interaction free from the spatiotemporal boundaries of the material world. In this sense, individuals, particularly youth, actively utilize digital media to construct their social relations (Ito et al. 2010). The participatory and interactive nature of digital media facilitates a "peer-based learning" process among youth, making it an essential element in their identity formation (Ito et al. 2010, pp. 21–23).

The online performances of hijabi YouTubers in Turkey also bear a close resemblance to the digital performances of other hijabi YouTubers worldwide. In this regard, veiling and the hijab are much more fashionable items in their styles, which reveal their close interaction with global trends via digital communication. At the same time, it is an outcome of the sociocultural transformation in Turkey, which began in the 1980s—with the adoption

of the neoliberal economic model—and accelerated in the 2000s during the ruling era of the Justice and Development Party (Adalet ve Kalkınma Partisi, hereinafter the AKP).

The sociopolitical and economic development of the Islamic bourgeoisie indicates the sociocultural transformation in Turkey under the rule of the AKP, which prepared the ground for the incorporation of the Muslim population into the modern world and facilitated the utilization of digitalization for religious purposes. This process can be defined as the democratization and liberalization process in Turkey, during which a compromise has been achieved between Islamic values and the global world with the formation of Islamic modernity (Yavuz 2009). Being followed by a drastic transformation in sociopolitical and cultural context, neoliberal policies in Turkey were accomplished to build reconciliation between the Islamic population and consumer culture, contributing to "the construction of new Muslim subjectivities" (Gökariksel and Secor 2009, p. 13). These new Muslim subjectivities, becoming much more visible during the AKP era, have significantly reshaped the religious frame in Turkey. Göle (2002) identifies this development of the public visibility of Islam as the "normalization" and modernization process of Muslim identity:

> actors of Islam blend into modern urban spaces, use global communication networks, engage in public debates, follow consumption patterns, learn market rules, enter into secular time, get acquainted with values of individuation, professionalism, and consumerism, and reflect upon their new practices. (Göle 2002, p. 174)

As a marker of Muslim female identity, veiling and its implications were also transformed during this period. Until the 2000s, veiling in Turkey was generally associated with class differences and emphasized the non-elite lifestyles of religiously oriented people. In the early decades of the Islamic revival throughout the 70s and 80s, veiling simply meant covering the body and the hair, stated as an Islamic obligation for Muslim women. Regarding this, veiling represents Muslim women's social and political identities as followers of Islamic norms. During the 1990s, the headscarf that Muslim women wore to cover their hair transformed into a commodity, a fashionable item, as well as a signifier of sociocultural status. In the context of veiling fashion, neoliberal transformation in Turkey and the economic development of the conservative bourgeoisie contributed to the emergence of new market and consumption patterns, namely the "Islamic culture of consumption" addressing Muslim identities (Gökariksel and Secor 2009). The rise of the Islamic culture of consumption can be understood as "the adaptation and appropriation of neoliberal capitalism" with "the redefinition and transformation of Islamic practices and values" in Turkey (Gökariksel and Secor 2009, p. 34). The development of the Islamic fashion industry in Turkey and the increasing socio-economic mobilization of religiously oriented individuals are significant factors that promote and facilitate the visibility of young veiled (hijabi) Muslim women on digital platforms. It also reconfigures the utilization of digital platforms by these women, not merely as consumers but also as digital content makers.

### 5.2. Muslim Male YouTubers in Turkey

The popularization of digital technologies has not solely increased the visibility and display of young Muslim females on digital platforms. Muslim men have too found significant opportunities to express themselves and share their worldviews and ideas via digital platforms. In this context, many young male Muslims who share religious content on digital platforms have recently proliferated in Turkey, mostly embracing and displaying a mentorship role to guide and teach the youth on religious topics. The channels analyzed in this study were also selected based on their popularity and productivity on YouTube. Twenty videos from each channel have been examined, and the diversity of topics has been mainly considered in the selection process of YouTube videos posted in these channels.

5.2.1. Sözler Köşkü

Sözler Köşkü, İlim ve Kültür Derneği (Science and Culture Association) was founded in 2010 with the initiative of a group of university students in Bornova, İzmir, to deliver

Islamic knowledge to people as much as possible via digital channels. As stated in their official website, they define their primary mission as "explaining Islamic facts, making them understood, practiced and lived in everyday life" (Sözler Köşkü 2021a).

Increasing their YouTube presence with many other channels over time, their main channel, tagged Sözler Köşkü, has been active since 2011 and is followed by 2.63 million people. There are 450 videos on this channel, increasing with new videos weekly. They introduce their channel as the biggest Islamic YouTube channel in Turkey. Indeed, videos posted in this channel are introduced in ten different categories based on their content. Talks with Islamic scholars, street interviews, videos discussing in/compatibility of specific scientific facts with Islamic knowledge, conversations with people coming from different backgrounds (military officer, prisoners, and more), informal chats on trendy topics, videos about the Qur'an, and conversations with religious celebrities can be listed as some of them.

Furthermore, they have opened many other channels such as a channel for children and other YouTube channels addressing people worldwide. Videos in these channels are either translated or have subtitles in different languages, including English, Arabic, Spanish, German. Sözler Köşkü also has its own Instagram, Twitter, and Facebook pages. In addition, some of the YouTubers such as Fatih Yağcı and Ceyhun Hamid Yılmaz—actors related to Sözler Köşkü—have their own YouTube channels in which they also post videos regarding Islamic topics. The content shared and discussed in most of these videos can be summarized as discussions, debates, and talks about Islamic knowledge and discussions of various topics within Islamic discourse. For example, in some videos, they present arguments to prove the existence of Allah, or they post Q&A videos with non-Muslims featuring discussions and debates about the "inevitable existence" of God. They also post videos providing arguments that refute some of scientific facts (e.g., Darwin's evolution theory) that are commonly understood and accepted as incompatible with Islam (Sözler Köşkü 2019a).

Tensions have arisen between different channels around these topics as well. For example, as a response to the video by Sözler Köşkü posted against the validity of Darwin's theory, Evrim Ağacı (Evolution Tree)—a scientific association—shared another YouTube video. In this video, a member of Evrim Ağacı explained that the arguments of Sözler Köşkü are based on logical fallacies and unscientific statements and could not be considered as sound arguments against the evolution theory (Evrim Ağacı 2019).

### 5.2.2. Hayalhanem

Hayalhanem is another Islamic YouTube channel that recently became popular in Turkey. The channel owner Mehmet Yıldız is a well-known YouTuber and author. He first joined YouTube with his channel Hayalhanem Istanbul in 2011, with 81,200 subscribers and 138 videos. His most popular channel is Hayalhanem Mersin, active on YouTube since May 2013 with 2.35 million followers and 754 videos, in addition to his personal YouTube channel, Facebook, Instagram, and Twitter accounts. Like Sözler Köşkü, Mehmet Yıldız is also very active on YouTube. As stated on its homepage, three new videos are added weekly to Hayalhanem Mersin.

Interestingly, Mehmet Yıldız did not have an educational background in theology or Islam in particular. He graduated from the mathematics department and received his master's degree from Muğla Sıtkı Koçman University; the department is unspecified (Kimnereli n.d.). Again, like Sözler Köşkü, the content of the videos in Hayalhanem is mostly related to Islamic issues. Mehmet Yıldız also shares videos about his activities with his fellows (his friends and the young males he is mentoring), such as their holiday videos or how they spent a fun day together (Hayalhanem 2018b). As we can understand from the comments, Mehmet Yıldız is also being followed by many women, while his fellows, as we can see from these videos, are strictly male, particularly young men.

### 5.2.3. The Road from YouTube to Islam: Young Male Muslim YouTubers as New Religious Actors

Mehmet Yıldız and Sözler Köşkü have become popular among religious youth in Turkey thanks to their visual and discursive styles. In this respect, both Sözler Köşkü and Hayalhanem differ from mainstream religious authorities by their alternative religious figure image.

One of the reasons for the popularity of these channels is the content they share in these videos. By discussing popular topics with an Islamic interpretation, posting either sensational or emotional titles to their videos, and making them more entertaining in an informal manner, these YouTubers can quickly draw the attention of young individuals. For example, one of the latest videos of Sözler Köşkü was about Squid Game, a top-rated Netflix series (Sözler Köşkü 2021b). All these topics are covered and discussed around an Islamic discourse concerning moral values, concluding with the suggestion of the spiritual satisfaction that people can gain by living according to Islamic values and practices. By covering different topics from an Islamic point of view, using a style of discourse that can be more attractive for youth, and being closer to the age of their followers, these young religious actors become very popular on digital platforms.

The words they choose and the jokes they crack make their speeches more entertaining and attractive for religious youth. Topics they discuss to deliver their messages also increase their popularity among youth. In this respect, we can see that they post many videos covering topics about intimate relationships between men and women, sexuality, and inspiring conversations on how to live a religious life without isolating themselves from the modern world. Their visual images and clothing styles also follow modern trends like their discursive styles. Although this does not mean that mainstream religious authorities do not mention these issues or that they do not have young followers, the popular discursive style and modern images of these young males combined with their effective utilization of digital media enable them to reach a wide audience. In this respect, their young followers can much more quickly identify themselves with these YouTubers than with elder authority figures.

Strikingly, although all these male figures have a modern look, their religious approach strictly follows mainstream Islamic discourse, even sometimes much more rigid than the views of elder and well-known religious authorities. The difference between the interpretation of some sexual practices by Cübbeli Ahmet Hoca—a famous religious actor and the leader of İsmailağa Jamaat—and by Mehmet Aydın or Sözler Köşkü exemplifies this situation. These male YouTubers state that according to Islamic norms, Muslims must refrain from sexual practices such as masturbation or oral sex since they are sinful activities (Hayalhanem 2015; Sözler Köşkü 2017). On the other hand, Cübbeli Ahmet Hoca interprets these sexual practices as inappropriate and sinful that Muslims should avoid as much as possible. However, he also adds that they can be permitted in specific cases to abstain from committing sins such as adultery, which ranks among the big sins (Cübbeli Ahmet Hoca 2019).

The free and democratic environment of digital media, which allows the generation and transmission of different ideas and interpretations, promotes the emergence of new religious actors such as these YouTubers. These YouTubers are not Islamic scholars, nor have they received any formal Islamic knowledge. Nevertheless, they present a mentor role guiding their audience on religion in their online speeches. They act as religious actors referring to Qur'an, hadiths, and several other religious resources. Many disputes and tensions have also been known to arise between certain religious authorities—mostly older and mainstream—and these YouTubers. Young male Muslim YouTubers are mostly criticized for introducing themselves as religious actors through their digital performances. Although most of their interpretations and Islamic conversations follow the mainstream Islamic perspective, some interpretations might conflict with different perspectives. In these cases, certain religious authorities can denounce them for their distorted (mis)interpretations (Çevik 2020).

The mediatization of these YouTubers is also not welcomed by mainstream religious authorities. Some religious authorities criticize modern or more casual images of these figures and their discursive styles for being too informal and eroding the sacred image of Islam. In this sense, we can see that the main reason for the dispute is the "banalization" effect of mediatization, which transforms and alters the way of understanding religion (Han 2016, p. 9). As religion is much more interpreted in popular or daily discourse, its sacredness diminishes as it more and more becomes a part of daily life. Although YouTube channels such as Sözler Köşkü and Hayalhanem—and other Islamic YouTube channels—aim to deliver the message of Islam to as many people as possible, their way of preaching follows the norms of media. In this regard, these young male YouTubers act as both religious actors and media celebrities. While they identify themselves as religious actors following Islamic norms and doctrines, their self-representation reflects the impact of digital popular culture in their formation of identity process. While they are more reluctant to share any posts relating to their family life, their content about their holiday trips with their friends or moments from their informal meetings also contributes to their self-promotion as micro-celebrities.

## 6. Conclusions

New media in the contemporary world differs from mainstream media channels by its decentralized and less regulated nature. Unlike mass media, digital media technologies offer people a virtual space "where people can democratically express themselves without artificial constraint and hindrance" (Possamai and Turner 2012, p. 198). The alternative space provided by new media technologies—particularly social media and digital platforms—offers users greater freedom for self-expression while intensifying social interaction. The so-called "democratic" environment of digital media provides individuals new opportunities for discussion and debate through its potential to expand social connectivity and promotes interactive, mass participatory, and dynamic communication. Cyberspace is also "globalizing" by its nature. It enables interactions free from the spatiotemporal constraints of the real world. The social interactivity and interconnectivity provided by digital media technologies open new fields where ideas, values, and practices can be shared, discussed, and contested, simultaneously promoting new ideas, meanings, and practices.

Not only Muslim youth but Muslim individuals in general, through social interaction and communication with different communities and encountering different views and experiences, can question their faith and discuss religious topics which have diverse effects on different people. While the online performances of hijabi women can inspire and encourage young Muslims to wear a hijab or find moral assistance through different blogs, others who do not choose this option can reach alternative views and interpretations that support and validate their decision (Akou 2010, p. 344).

In the utilization of digital platforms by Muslim YouTubers in Turkey, it can be easily inferred that their ways of self-representation are inevitably being influenced by globalization, digital culture, and modern values while reflecting the impact of the sociopolitical structures they live in. Their visual images and discourse styles represent an amalgamation of modernity, popular culture, and Islamic norms, and their digital performances are reshaped "around the themes of self-realization, personal autonomy, and emotional expressivity" (Turner 2012, p. 139). On the other hand, significant differences appear between the contents of male and female Muslim YouTubers in Turkey. Their expression of their religiosity mainly differs in terms of style and manner. Here, the difference between their internalization of modern culture and its expression is at stake. Hijabi YouTubers mostly follow popular trends to prioritize the harmony and compatibility between modern lifestyles and their religious personalities. In other words, their self-representation draws the image of modern Muslim women. On the contrary, despite having a modern look and following the norms of media, young male YouTubers heavily criticize modern culture and modern lifestyles. While following global trends and drawing a portrait of a modernized Muslim man, in their speeches and videos, they constantly caution their followers about

the threats and dangers of the modern world that might distract young individuals from the Islamic way of life (Hayalhanem 2018a).

Female and male Muslim YouTubers in Turkey also diverge from each other in their different utilizations of digital platforms. Following Schiffer's categorization of artifacts, considering these platforms as digital objects, we can categorize three different functionalities: "techno function (the utilitarian function of the object); sociofunction (impact on social interaction and communication, social representation); ideofunction (symbolization of ideas, values, etc.)" (Schiffer 1992; Bağlı 2015). These three functions are mostly intertwined with each other and can be experienced simultaneously within the utilization of digital objects as artifacts. In this respect, the utilitarian function of YouTube is recognizably clear in its utilization by both male and female Muslim YouTubers. Furthermore, besides expressing themselves and reaching out to more people, both YouTubers make money as content-makers/producers and influencers.

However, regarding the content of hijabi YouTubers, we can argue that they tend to make use of and prioritize socialization and networking in their utilization of YouTube. They choose to produce contents that highly represent and express their individuality (fashion blogs, vlogs, etc.), which promotes social interaction, connectivity, and communication among their online community (i.e., friends, followers, etc.). In this respect, individuality and socialization become prominent in the YouTube content created by hijabi YouTubers in Turkey. Rather than establishing an identity based on their religious personae and acting as religious figures, these women choose to follow global trends. Through their online self-representation, they draw the image of liberated Muslim women who successfully combine their religious personality with modern values. More importantly, even though they do not claim any religious statement, their "alternative" styles, their self-representation challenge the normative Islamic discourse, mainly based on a patriarchal interpretation of Islam.

On the contrary, although more reluctant to share their individual experiences, Muslim male YouTubers tend to use digital media for delivering the Islamic message to as many people as possible. In this respect, they utilize digital media as an extension of their appearance in society as religious actors, and the ideofunction is prioritized in their utilization of YouTube. Furthermore, YouTube content produced by Muslim male YouTubers reflects their ideological position, which is based on patriarchal norms. In this context, the mentorship role male YouTubers assign themselves and their speeches highlights the significant role of gender-based hierarchy in shaping social structures and even in the process of the identity formation of young individuals. While underlining the significance of women in Islam, Muslim male YouTubers also reject gender equality between men and women. As can be followed from their videos, they draw their arguments from the patriarchal definition of women and define them as vulnerable and emotional subjects by nature that should be protected by men (Hayalhanem 2016; Sözler Köşkü 2019b). They also discuss the issue of veiling widely in many videos, advocating that veiling is a requirement that Muslim women must fulfill.

It is also crucial to note that the increased utilization of social media and digital platforms offers young people new opportunities to express their self-identities with more complex, diverse, and subjective definitions that challenge normative discourse and mainstream categories. Hence, practices and self-definition of identities in digital media platforms imply "a proliferation of identity labels and practices that help accommodate complexity, intersectionality and fluidity" (Cover 2018). In this respect, identity formation is a dialectical process that has been constantly reshaped and altered under the impact of social structures. While these YouTubers represent themselves as Muslim individuals, their self-representation reflects the impact of the main features of contemporary societies such as globalization, consumerism, and digital culture.

The generation of alternative views, interpretations, and the visibility of different online identities facilitate the democratization of religion and make it more compatible with modernity. Opening space for counter positions, digital media enables the emergence of

different religious figures and allows for the expression of contested perspectives, interpretations, and ideas via digital communication. In this context, perhaps not religion, but ways of living religion, personal experiences, and the manifestation of religious perspectives—in other words, religiosity—have been constantly reformulated and reshaped, particularly with the impact of religious activities online and religious figures online.

**Funding:** This research received no external funding.

**Institutional Review Board Statement:** Not applicable.

**Informed Consent Statement:** Not applicable.

**Data Availability Statement:** Not applicable.

**Conflicts of Interest:** The author declares no conflict of interest.

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
