# Peer review of "Muslim YouTubers in Turkey and the Authoritarian Male Gaze on YouTube"

_religions, doi:10.3390/rel13040318_

Round 1

Reviewer 1 Report

I enjoyed reading the article; it was interesting and had a good flow. It also draws upon significant issues regarding performativity, alternative lifestyles and religious way of living in the so-called "modernized sphere" of digital media. As always, there are a few aspects to consider for a better and more consistent read: 

  • The author refers (on pg. 1, intro) to the impact of digital media technologies on netizens (self identification, self perception, self positioning in society and the global world)- needs reference (i.e. M. Cooke, F. Liu)
  • In the methods part, it would be useful to describe the reasons behind the selection of the sample. Is it the themes/content, popularity, number of followers, feeds etc. Is there a comparative intention behind this as well?  
  • Again, in the methods part: The author mentions exploring different utilization of YouTube via virtual ethnography and micro-analysis (it would be worthwhile to mention visual content and discourse analysis here to expand on the micro-analysis. Then it makes more sense to the reader.) 
  • On pg. 6: "...but digital media also make visible and provide a new form of social infrastructure for the individual’s religion: a network of local communities” (Lövheim 263 2016, p. 52). I suggest reflecting and expanding on this phenomenon since it builds upon the new modernisms/modern ways of religious living hosted by digital media, introduced earlier in the paper. Ito`s work would be useful in that regard. 
  • On pg. 9: Easy transmission of alternative views and ideas in online communities and their accessibility also leads to a decline in the significance of traditional religious authorities.... (this statement is based on prediction and must be removed or supported with evidence) 
  • On pg 11 line 541: "In this context, the authoritarian role..." (here the author probably means the mentor role and the authority figure. In its present form, it has an ideological connotation).
  • For the intricate relationship between religion, gender, digitalization, globalization, and culture and more specifically identity construction, globalization and the impact of digital media; I would suggest looking into further literature: More recent would be Farrukh et al: https://onlinelibrary.wiley.com/doi/abs/10.1111/sena.12344;  earlier work of Ito, Coleman and Liu are also focused on the topic.

Author Response

Thank you very much for your kind comments and suggestions. Please see the attachment.

Reviewer 2 Report

The paper's goals are laudatory and the paper has the potential to make an important contribution to the literature.  In its current state, however, the paper lacks clarity and overstates its "findings" given the methodology employed.

Suggestions:

1. The title should be specific to Turkey and not suggest generalizability to all Muslims.

2.  There are a number of grammatical problems and typos, e.g., Line 12: in what ways does digital culture alters and shapes self-representations [should be alter]

3.  The author does not introduce citations until line 49. This strikes me as very odd in terms of standards for lit reviews in published papers, as earlier on in the paper, the author needed to support various statements.  

4.  The paper was hard to understand. I had to pause after each sentence to figure out the intended meaning. 

In addition, the author needs to avoid words that convey a value judgment:  line 73: “Ground on dogmatic principles” (which to be idiomatic should actually be "grounded in dogmatic . . . .").

"Dogmatic" may be the intended idea, i.e., "inclined to lay down principles as incontrovertibly true'", however, the word has negative connotations that could be deemed offensive.  If the author is instead trying to convey  how "dogmatic theology emphasizes the importance of propositional truth over experiential, sensory perceptions" then that should be expressed in a way that is not disparaging. 

Lines 73-9 as a section exemplify how the paper is written in a way that is confusing and awkward:

“Despite conceptually being grounded on dogmatic principles, and having been influenced by past knowledge and experiences, ways of how religion is understood and practiced are always already in interaction with the sociopolitical and even material conditions of the time and place in which they are performed. Digitalization and technology, being the main facets of the modern world, have a recognizable effect on religious understanding, rituals, and the manifestation of religiosity in today’s world. “

5.  The paper's goal is not clearly articulated.  After reading the whole paper, I searched for what the paper was trying to accomplish in the Abstract, line 16+:

". . . sociofunctionality and individuality are much more prioritized in the contents created by female Muslim YouTubers, while an ideological and authoritarian perspective becomes prominent among the YouTube videos created by male Muslim YouTubers in Turkey."

I then searched for this main point in the paper:  "Hijabi YouTubers mostly follow popular trends to prioritize the harmony and compatibility between modern lifestyles and their religious personality, in other words, their self-representation draws the image of modern Muslim women. On the contrary, despite having a modern look and following the norms of media, young male YouTubers heavily criticize the modern culture and modern lifestyles."

6.  Once I finally understood the gist of the paper, I went back to see if the methodological could reasonably support such a claim.  Unfortunately, I do not believe it can.

The author says:

Methods:

"I will first examine three female Muslim YouTubers, respectively, Rimel askina, Aslı Afşaroğlu, and Sule Celik. Then, I will analyze two Islamic YouTube channels, designed by young male Muslims in Turkey, respectively Sözler Köşkü (a channel organized by a group of Muslim males) and Hayalhanem (Mehmet Aydın)."

This is a tiny sample and we have no way of knowing if most of the programming in Turkey of this type is presented by these few channels/performers, or if those studied are just a fraction of all the similar programming available in Turkey. 

How/why were these persons selected for the study?

In addition, I would have liked to have a better understanding of how the author systematically examined the content to derive themes on which the conclusions are based.  A table or chart would have been helpful so that a comparison by gender would be clearer.

In sum, there seems to be interesting material buried in this paper but its content needs clearer contextualization from the literature, and much more information about the sampling and analysis of performers studied. 

Despite attention to these suggested improvements, I'm not sure that the author would be able to convince readers of the validity of the main points with such a small sample. Furthermore, discussion of the limitations of the paper was conspicuously absent. 

Finally, a significant impediment was the language which seemed convoluted.  The author might consider eliminating jargon such as "sociofunctionality".  Clarity should be prioritized over words that attempt to encapsulate concepts that should simply be articulated in a way that is accessible to a maximal number of readers. 

Best of luck with revisions as this is an area worthy of study. 

Reviewer 3 Report

The article is very well written with a clear and interesting introduction. 

The research questions do not appear explicitly, which makes it hard to decide if they are answered. But the conclusion and results and discussion lifts coherent and interesting material, analyses and points.

Methodology:

Micro-analyses should be explained, what is it?

Chapter 4.1should be  a little bit more specific regarding the selection

Line 196 perhaps "introduce" rather than "mention"

Author Response

Thank you very much for your nice comments. Please see the attachment.

Round 2

Reviewer 2 Report

The author has engaged in extensive revisions that have addressed my concerns.  I applaud the author for responding so appropriately to my suggestions.  

Author Response

Thank you very much.